# Effect of dietary inclusion of artichoke (*Cynara scolymus*) leaf powder and astaxanthin on productive performance, hematological and biochemical profiles, hepatic and antioxidant enzyme activities, and gut microbiota in broiler chickens

Saad Naji Nasser[1], Mehran Torki[1]*, Fadhil Rasoul Abbas Al-Khafaji[2], Shahab Ghazi Harsini[1], Ali Ahmad Alaw Qotbi[2]

**1** Animal Science Department, College of Agriculture and Natural Resources, Razi University, Kermanshah, Iran, **2** Animal Production Department, College of Agriculture, Al-Qasim Green University, Babylon, Iraq

* torki@razi.ac.ir

## Abstract

Recognizing the growing demand for sustainable and natural alternatives in poultry production, this study evaluated the effects of dietary inclusion of artichoke (*Cynara scolymus*) leaf powder (ALP) and astaxanthin on growth performance, hematological and biochemical profiles, hepatic and antioxidant enzyme activities, and gut microbiota in broiler chickens. A total of 1,080 one-day-old Ross 308 broiler chicks were randomly assigned to nine dietary treatments (six replicates each) for 35 days: a control diet (T1) received basal diet without additives; T2 and T3: 60 and 120 mg astaxanthin/kg feed; T4 and T5: 1 and 2 g ALP/kg feed; T6: 60 mg astaxanthin + 1 g ALP/kg feed; T7: 60 mg astaxanthin + 2 g ALP/kg feed; T8: 120 mg astaxanthin + 1 g ALP/kg feed; T9: 120 mg astaxanthin + 2 g ALP/kg feed. The results showed that the groups fed T2 and T6 significantly enhanced growth performance, increasing live body weight (LBW, p < 0.01) and body weight gain (BWG, p < 0.01) while reducing feed conversion ratio (FCR, p < 0.01). The groups receiving T2, T6, and T9 lowered heterophil counts and heterophil/lymphocyte (H/L) ratios (p < 0.01). The T4 group reduced cholesterol (p < 0.01), while groups receiving T4, T5, T8, and T9 decreased uric acid levels (p < 0.01). Total protein increased in the T5 and T6 groups (p < 0.01). Antioxidant enzyme activities improved, with elevated glutathione peroxidase (GPX) in all supplemented groups, particularly T9 group (p < 0.01), and increased catalase (CAT) in broilers fed T6 and T7 (p < 0.05). All supplemented groups reduced *E. coli* (p < 0.05) and increased *Lactobacillus* counts in the cecum (p < 0.01). In conclusion, astaxanthin and ALP supplementation, particularly 60 mg astaxanthin with 1 g ALP (T6), enhanced broiler growth, antioxidant status, metabolic health, and gut microbiota.

**Data availability statement:** All relevant data are within the paper and its Supporting Information files.

**Funding:** The author(s) received no specific funding for this work.

**Competing interests:** The authors have declared that no competing interests exist.

## Introduction

Globally, poultry meat has become a primary source of high-quality animal protein, addressing the nutritional needs of a burgeoning population [1]. To meet this escalating demand, the poultry industry has intensified research and development efforts to optimize production efficiency and bolster bird health. However, commercial poultry production is subject to various environmental, technological, nutritional, and biological/internal stressors that can adversely affect productivity, reproductive performance, and overall health [2,3]. Recent studies have underscored that the overproduction of free radicals, coupled with compromised antioxidant defenses and oxidative stress, plays a significant role in exacerbating the negative impacts of these stressors on poultry [2]. Furthermore, the prohibition of antibiotics as growth promoters has increased the risk of disease within poultry populations, further contributing to oxidative stress [3]. In this context, natural antioxidants such as carotenoids and polyphenols have gained significant attention as promising dietary supplements to alleviate oxidative stress and promote the health and productivity of poultry.

Astaxanthin, a xanthophyll carotenoid characterized by its distinctive polyene chain and polar end groups, has been recognized for its potent antioxidant properties [4]. This natural pigment, sourced primarily from algae and yeast, efficiently scavenges reactive oxygen species (ROS) and integrates into cell membranes, providing comprehensive antioxidant protection [4]. *Haematococcus pluvialis* (*H. pluvialis*), a unicellular microalga commonly found in various environments, is a widely utilized as primary source organism for astaxanthin production [5]. This carotenoid pigment can accumulate up to 5% of the algae's dry weight, making *H. pluvialis* a valuable commercial source [4]. Recent research has underscored the beneficial effects of astaxanthin supplementation on poultry health and performance [6,7].

The artichoke (*Cynara scolymus L.*) is a versatile plant with applications spanning culinary, pharmaceutical, and animal feed industries. Renowned for its antioxidant properties, artichoke is a rich source of phenolic compounds, including cynarin, a potent antioxidant [8]. Additionally, it contains a diverse array of flavonoids, such as apigenin, luteolin, cyanidin, peonidin, and delphinidin, which have been shown to effectively scavenge ROS and other free radicals [8]. Beyond its antioxidant potential, artichoke is a nutritional powerhouse, rich in vitamins, minerals, and dietary fiber [9]. It is also a significant source of prebiotic fibers, such as fructans, inulin, and oligofructose, which promote gut health [9]. Other beneficial compounds found in artichoke include tannins and organic acids, contributing to its medicinal, antibiotic, and antioxidant properties [10]. Previous studies have reported increased feed intake (FI), body weight gain (BWG), and decreased feed conversion ratio (FCR) in poultry fed artichoke-based diets [11,12]. By investigating the combined effects of astaxanthin and ALP on growth performance, hematological and biochemical profiles, hepatic and antioxidant enzyme activities, and gut microbiota, this study aims to provide valuable insights into the potential of these natural supplements to enhance broiler chicken health and productivity.

## Materials and methods

### Ethics statement

This experiment was authorized by the Razi University Animal Ethics Committee (Kermanshah, Iran; IR.RAZI. REC.1403.008). All experimental procedures were conducted in adherence to the European Union's stringent animal welfare and feed safety regulations.

### Experimental design, birds, management, and diets

The study was performed at the research facility of Al-Anwar Poultry Company in Babel Governorate, Iraq. A total of 1080 unsexed Ross 308 one-day-old broiler chicks, weighing an average of 37–38 g, were randomly allocated to nine dietary treatments. Each treatment group consisted of six replicates of 20 chicks. The experimental period lasted for 35 days. The nine dietary treatments were as follows: T1: Control: Standard diet without additives; T2: astaxanthin 60 mg/kg; T3: astaxanthin 120 mg/kg; T4: ALP 1 g/kg; T5: ALP 2 g/kg; T6: astaxanthin 60 mg/kg+ALP 1 g/kg; T7: astaxanthin 60 mg/kg+ALP 2 g/kg; T8: astaxanthin 120 mg/kg+ALP 1 g/kg; T9: astaxanthin 120 mg/kg+ALP 2 g/kg. Feed and water were provided *ad libitum* throughout the experiment. The composition of the starter and grower diets is presented in Table 1. Astaxanthin powder, derived from the algae *H. pluvialis* and sourced from AstaPure Company, and ALP, obtained from Prescribed for Life, were incorporated into the diets from the first day of the experiment. The chicks were housed in ground cages

**Table 1.** Ingredients and composition of the experimental diets.

| | Starter diet | Grower diet |
|---|---|---|
| Ingredient (%) | 1-21 days | 22-35 days |
| Corn | 30.00 | 40.00 |
| Wheat | 28.25 | 24.00 |
| Soybean meal (48% protein) | 31.75 | 24.8 |
| Protein concentrate[1] | 5.00 | 5.00 |
| Sunflower oil | 2.9 | 4.4 |
| Limestone | 0.9 | 0.6 |
| Dicalcium phosphate (DCP) | 0.7 | 0.9 |
| Mixture of vitamins and minerals[2] | 0.2 | 0.2 |
| NaCl | 0.3 | 0.1 |
| Chemical composition | | |
| Crude protein (%) | 23.04 | 20.06 |
| Metabolizable energy (kcal/kg feed) | 3021.45 | 3194.92 |
| Lysine % | 1.27 | 1.07 |
| Methionine % | 0.41 | 0.38 |
| Cysteine % | 0.35 | 0.30 |
| Methionine+Cysteine % | 0.82 | 0.78 |
| Available phosphorus % | 0.41 | 0.43 |
| Energy: Protein % | 131.14 | 159.27 |

[1]The protein concentrate used was Brocon-5 Special W (Chinese origin). Per kg, it contained: 40% crude protein, 3.5% fat, 1% fiber, 6% calcium, 3% available phosphorus, 3.25% lysine, 3.90% methionine+cysteine, 2.2% sodium, 2100 kcal/kg represented energy.

[2]Each kg contained: 20,000 IU Vitamin A, 40,000 IU Vitamin D3, 500 mg Vitamin E, 30 mg Vitamin K3, 15 mg Vitamin B1+B2, 150 mg B3, 20 mg B6, 300 mg B12, 10 mg folic acid, 100 µg biotin, 1 mg iron, 100 mg copper, 1.2 mg manganese, 800 mg zinc, 15 mg iodine, 2 mg selenium, 6 mg cobalt, and 900 mg antioxidants (BHT).

measuring 1 × 1.5 × 1 m, with sawdust bedding. Temperature and humidity were controlled using gas incubators and air evacuators. A standard vaccination program was implemented against infectious bronchitis, Newcastle disease, infectious bursal disease, and avian influenza.

## Sampling and measurements

**Growth performance.** Broiler performance was assessed through weekly measurements of live body weight (LBW), BWG, FI, and FCR. For all parameters, the mean value over 5 consecutive weeks was considered. LBW was determined by weekly individual bird weighing, and BWG was calculated as the change in LBW between successive measurements. FI was calculated by subtracting residual feed from the initial feed provided, then dividing by the number of birds per group. The FCR was subsequently derived as the quotient of weekly FI (g) and weekly BWG (g).

**Hematological and biochemical parameters.** At the end of the 35-day study, blood samples were collected from the wing vein of one broilers per replicate. Anticoagulant-treated tubes were used to obtain blood for the assessment of hematological profiles, which included: red and white blood cell counts (RBC and WBC), packed cell volume (PCV, %), hemoglobin concentration (Hb) [13], heterophil and lymphocyte percentages, and the heterophil-to-lymphocyte (H/L) ratio [14]. To determine serum biochemical profiles, additional blood samples were collected into tubes devoid of anticoagulant. Serum was separated through centrifugation at 3000 rpm for 15 minutes and subsequently stored at −20°C until analysis. Serum glucose, cholesterol, uric acid, and total protein concentrations were measured using commercially available kits (Pars Azmun, Tehran, Iran).

**Hepatic and antioxidant enzyme activities assays.** Blood serum enzyme activities were quantified using commercially available assay kits. Alanine aminotransferase (ALT) and aspartate aminotransferase (AST) levels were determined using kits from Orphee (France), adhering to the manufacturer's instructions. Glutathione peroxidase (GPX) activity was measured following the procedure described in EFSA [15]. Catalase (CAT) activity was assessed using a kit from Orphee (France), based on the method detailed by Ambati et al [16].

**Microbiota measurements.** To assess cecal microbial populations, one broilers per pen were subjected to aseptic euthanasia at the experiment's conclusion. Cecal digesta was immediately collected and processed in a laboratory setting. A serial tenfold dilution series was initiated by combining one gram of cecal material with nine milliliters of sterile physiological saline (0.9% NaCl), culminating in a $10^{-5}$ dilution. *Lactobacillus* spp., *E. coli*, and total aerobic bacteria were quantified using MRS agar, MacConkey agar, and nutrient agar (NA), respectively (all media from Merck, Germany). 250 μL aliquots from each dilution were cultured on the designated media and incubated at 37°C for 48 hours (MRS) or 24 hours (MacConkey and NA). Post-incubation, colony counts were determined, and the bacterial load, expressed as CFU per gram of cecal content, was calculated by applying the appropriate dilution factor. All microbiological manipulations, including media preparation, sample plating, incubation, and colony counting, were conducted within a sterile environment provided by a microbiological safety cabinet.

## Statistical analysis

To evaluate the effects of experimental treatments on the measured traits, a completely randomized design (CRD) was utilized. Statistical analysis was performed using SAS 9.4 software [17]. The following statistical model was utilized to assess treatment effects: $Y_{ij} = \mu + T_i + e_{ij}$.

In this model, $Y_{ij}$ is the observed response for the jth replicate of the ith treatment, $\mu$ is the overall mean, $T_i$ is the effect of the ith treatment (i = 1–9), and $e_{ij}$ is the random error term. Significant differences among treatment means were determined using Duncan's multiple range test [18] ($p < 0.01$ and $p < 0.05$). All data are presented as means ± standard errors of the means.

## Results

### Growth performance

Table 2 summarizes the impact of varying levels of astaxanthin, ALP, and their combination on the growth performance (LBW, BWG, FI, and FCR) of broilers. Across the entire experimental period the highest overall LBW and BWG were observed in groups fed with T6 and T2, which showed a significant difference compared to all other experimental treatments ($p < 0.01$). Conversely, the lowest LBW and BWG were recorded in broilers under the control group, exhibiting a significant difference from the other experimental treatments ($p < 0.01$). Regarding FI, the highest consumption was observed in groups fed with T4, followed by T2 ($p < 0.01$). The lowest FI was recorded sequentially in groups fed with T8, T1, T5, and finally T9 ($p < 0.01$). In terms of FCR, the maximum value was recorded in the group administered T4, while the minimum FCR was achieved in the group receiving T6, followed closely by the group on T8 ($p < 0.01$). The excel file of performance data is presented as S1 Data.

### Hematological parameters

Table 3 summarizes the impact of varying levels of astaxanthin, ALP, and their combination on broiler hematological parameters. The treatments significantly influenced RBC count, PCV, Hb, heterophil and lymphocyte numbers, and the H/L ratio ($p < 0.05$). RBC counts varied across treatments, with the groups receiving T4 and T8 showing the highest counts, while the group receiving T6 exhibited the lowest ($p < 0.05$).

Regarding PCV and Hb concentrations, the highest values were recorded in broilers provided with T9, while the lowest were observed in broilers under T7 ($p < 0.05$); however, no differences were found among the other treatments.

Heterophil counts showed the greatest increase in broilers fed T8, while T2, T6, and T9 groups resulted in lower heterophil counts than the control group ($p < 0.01$). Moreover, lymphocyte counts were lowest in broilers receiving T4, and higher in T1, T2, T3, and T7 groups ($p < 0.05$).

The H/L ratio was highest in broilers consuming T8 and lowest in those supplemented with T2, both differing from the control ($p < 0.01$). In contrast, WBC counts remained unaffected by the experimental treatments ($p > 0.05$).

**Table 2. Impact of astaxanthin and artichoke leaf powder on the growth performance of broilers.**

| Treatments | Initial weight (g) | Live body weight (g) | Body weight gain (g) | Feed intake (g) | Feed conversion ratio (g/g) |
|---|---|---|---|---|---|
| T1 | 37.20 ± 0.13 | 1986.53 ± 1.76[d] | 1949.33 ± 1.69[d] | 2778.06 ± 0.72[g] | 1.42 ± 0.00[b] |
| T2 | 37.78 ± 0.07 | 2101.48 ± 2.46[a] | 2063.71 ± 2.44[a] | 2867.94 ± 2.05[b] | 1.38 ± 0.00[d] |
| T3 | 37.05 ± 0.05 | 2055.23 ± 4.49[b] | 2018.19 ± 4.52[b] | 2827.09 ± 1.49[c] | 1.39 ± 0.00[c] |
| T4 | 37.60 ± 0.08 | 2004.37 ± 0.53[c] | 1966.77 ± 0.50[c] | 2880.71 ± 3.36[a] | 1.45 ± 0.00[a] |
| T5 | 37.81 ± 0.08 | 2054.31 ± 0.96[b] | 2016.50 ± 0.91[b] | 2776.86 ± 0.55[g] | 1.37 ± 0.00[e] |
| T6 | 38.01 ± 0.10 | 2106.55 ± 1.29[a] | 2068.54 ± 1.34[a] | 2794.15 ± 1.52[e] | 1.35 ± 0.00[g] |
| T7 | 37.80 ± 0.07 | 2002.24 ± 0.43[c] | 1964.44 ± 0.43[c] | 2809.19 ± 0.62[d] | 1.42 ± 0.00[b] |
| T8 | 37.66 ± 0.05 | 2059.11 ± 4.85[b] | 2021.45 ± 4.85[b] | 2763.50 ± 0.73[h] | 1.36 ± 0.00[f] |
| T9 | 37.98 ± 0.01 | 1998.74 ± 0.38[c] | 1960.76 ± 0.39[c] | 2788.29 ± 0.41[f] | 1.42 ± 0.00[b] |
| P value | NS | ** | ** | ** | ** |

Values are mean ± standard error.

Treatments: T1 = Control treatment; T2 & T3 = 60 and 120 mg astaxanthin/ kg feed; T4 & T5 = 1 and 2 g artichoke leaf powder/ kg feed; T6 = 60 mg astaxanthin + 1 g artichoke leaf powder)/ kg feed; T7 = 60 mg astaxanthin + 2 g artichoke leaf powder/ kg feed; T8 = 120 mg astaxanthin + 1 g artichoke leaf powder/ kg feed; T9 = 120 mg astaxanthin + 2 g artichoke leaf powder/ kg feed.

Means within a column marked with different superscripts indicate statistically significant differences at **$p < 0.01$.

**Table 3. Impact of astaxanthin and artichoke leaf powder on broiler hematological parameters.**

| Treatments | RBC, 10⁶/mm³ | WBC, 10³/mm³ | PCV, % | Hb, g/100 ml | Heterophil % | Lymphocyte % | H/L |
|---|---|---|---|---|---|---|---|
| T1 | 3395.00±5.00$^{ab}$ | 30.50±0.50 | 28.50±0.50$^{ab}$ | 9.49±0.16$^{ab}$ | 21.00±1.00$^{b}$ | 65.50±0.50$^{a}$ | 0.31±0.01$^{b}$ |
| T2 | 3490.00±5.00$^{ab}$ | 31.25±0.25 | 28.50±0.50$^{ab}$ | 9.49±0.16$^{ab}$ | 13.50±0.50$^{d}$ | 65.00±1.00$^{a}$ | 0.20±0.00$^{d}$ |
| T3 | 3630.00±10.00$^{ab}$ | 32.00±1.00 | 28.50±0.50$^{ab}$ | 9.49±0.16$^{ab}$ | 19.50±0.50$^{bc}$ | 65.50±0.50$^{a}$ | 0.30±0.01$^{bc}$ |
| T4 | 3875.00±25.00$^{a}$ | 31.25±0.25 | 28.00±0.50$^{ab}$ | 9.33±0.16$^{ab}$ | 20.00±0.50$^{bc}$ | 62.00±1.00$^{b}$ | 0.31±0.00$^{b}$ |
| T5 | 3375.00±25.00$^{ab}$ | 32.20±0.20 | 28.50±0.50$^{ab}$ | 9.49±0.16$^{ab}$ | 21.00±1.00$^{b}$ | 63.00±1.00$^{ab}$ | 0.33±0.01$^{ab}$ |
| T6 | 3153.00±513.00$^{b}$ | 30.95±0.55 | 28.50±0.50$^{ab}$ | 9.49±0.16$^{ab}$ | 18.50±0.50$^{c}$ | 64.00±1.00$^{ab}$ | 0.28±0.00$^{c}$ |
| T7 | 3665.00±15.00$^{ab}$ | 30.50±0.50 | 27.50±0.50$^{b}$ | 9.16±0.16$^{b}$ | 20.50±0.50$^{bc}$ | 65.00±1.00$^{a}$ | 0.31±0.01$^{b}$ |
| T8 | 3850.00±50.00$^{a}$ | 31.50±0.50 | 28.50±0.50$^{ab}$ | 9.49±0.16$^{ab}$ | 23.50±0.50$^{a}$ | 64.50±0.50$^{ab}$ | 0.35±0.00$^{a}$ |
| T9 | 3438.50±6.50$^{ab}$ | 31.75±0.25 | 29.50±0.50$^{a}$ | 9.83±0.17$^{a}$ | 18.50±0.50$^{c}$ | 63.50±0.50$^{ab}$ | 0.29±0.01$^{c}$ |
| P value | * | NS | * | * | ** | * | ** |

Values are mean±standard error.

RBC: Red blood cell; WBC: White blood cell; PCV: Packed cell volume; Hb: Hemoglubin; H/L: Heterophil-to-lymphocyte ratio

Treatments: T1=Control treatment; T2 & T3=60 and 120 mg Astaxanthin/ kg feed; T4 & T5=1 and 2 g artichoke leaf powder/ kg feed; T6=60 mg Astaxanthin+1 g artichoke leaf powder)/ kg feed; T7=60 mg Astaxanthin+2 g artichoke leaf powder/ kg feed; T8=120 mg Astaxanthin+1 g artichoke leaf powder/ kg feed; T9=120 mg Astaxanthin+2 g artichoke leaf powder/ kg feed.

Means within a column marked with different superscripts indicate statistically significant differences at ** = (p<0.01), * = (p<0.05).

## Blood biochemical parameters

The effects of different levels of astaxanthin, ALP, and their combination on broiler blood biochemical characteristics are summarized in Figs 1 and 2. The experimental treatments had a significant impact on the concentrations of glucose, cholesterol, uric acid, and total protein (p<0.01). All experimental treatments, with the exception of T4, resulted in an increase in blood glucose concentration compared to the control group (p<0.01) (Fig 1).

Regarding the blood cholesterol concentration in broiler chickens, the lowest level was observed in broilers fed T4, while the highest levels were recorded in T3 and T8 groups (p<0.01) (Fig 1).

The lowest uric acid concentration was noted in the groups receiving T4, T8, T9 and T5, whereas the highest concentration was observed in T3 and T7 groups (p<0.01) (Fig 2).

In terms of total protein concentration, broilers under T5 and T6 exhibited the highest levels, which were different from most of the other treatments (p<0.01) (Fig 2). Conversely, the lowest total protein concentration was recorded in the group receiving T4.

## Hepatic and antioxidant enzyme activities

As shown in Figs 3–5, experimental treatments induced significant variations in hepatic and antioxidant enzyme levels, as measured by blood enzyme activities. Notably, ALT activity was maximized in broilers receiving T6 and minimized in those given T2 and T8 (p<0.01) (Fig 3). Conversely, AST activity peaked in broilers administered T2 and T7, significantly exceeding that of the T3 group (p<0.05) (Fig 4).

GPX activity was significantly depressed in the control group compared to all other treatments, with broilers on the T9 regimen exhibiting the highest activity (p<0.01) (Fig 5). CAT activity was significantly elevated in broilers provided with T6 and T7, with those in the T5 group also showing comparable levels, relative to the remaining treatments (p<0.05) (Fig 5).

## Caecum microbiota

The effects of experimental treatments on broiler caecal microbiota are presented in Fig 6. Significant variations in bacterial populations were observed across treatments. Notably, a consistent and significant decrease in *E. coli* populations

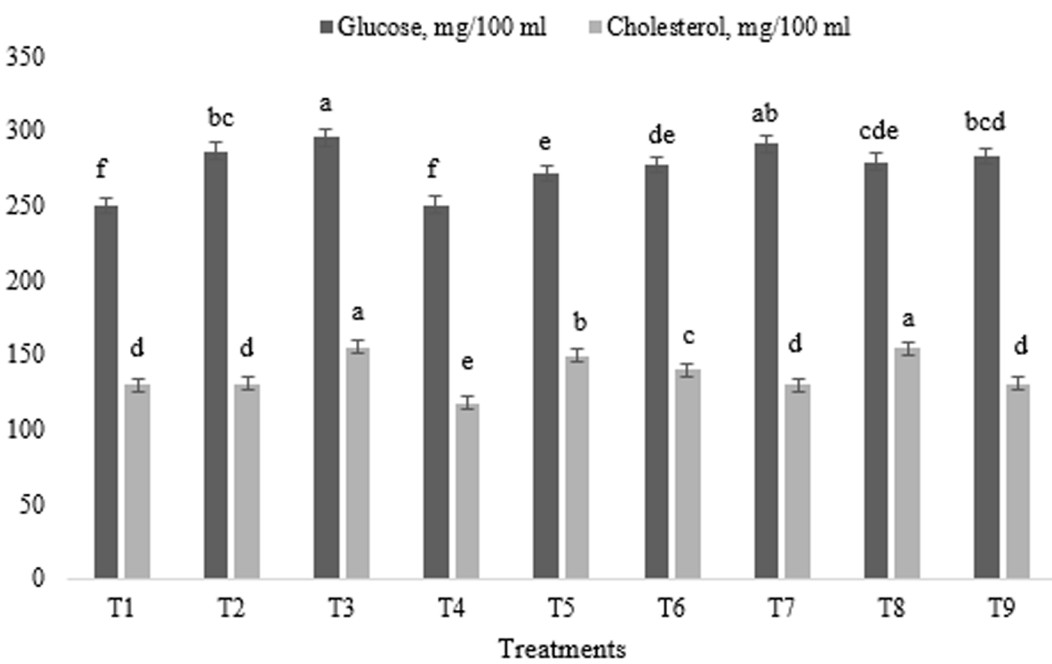

**Fig 1. Impact of astaxanthin and artichoke leaf powder on broiler blood glucose and cholesterol concentrations.** Treatments: T1 = Control treatment; T2 & T3 = 60 and 120 mg astaxanthin/ kg feed; T4 & T5 = 1 and 2 g artichoke leaf powder/ kg feed; T6 = 60 mg astaxanthin + 1 g artichoke leaf powder)/ kg feed; T7 = 60 mg astaxanthin + 2 g artichoke leaf powder/ kg feed; T8 = 120 mg astaxanthin + 1 g artichoke leaf powder/ kg feed; T9 = 120 mg astaxanthin + 2 g artichoke leaf powder/ kg feed. Means within a column marked with different superscripts indicate statistically significant differences $p < 0.01$ or $p < 0.05$.

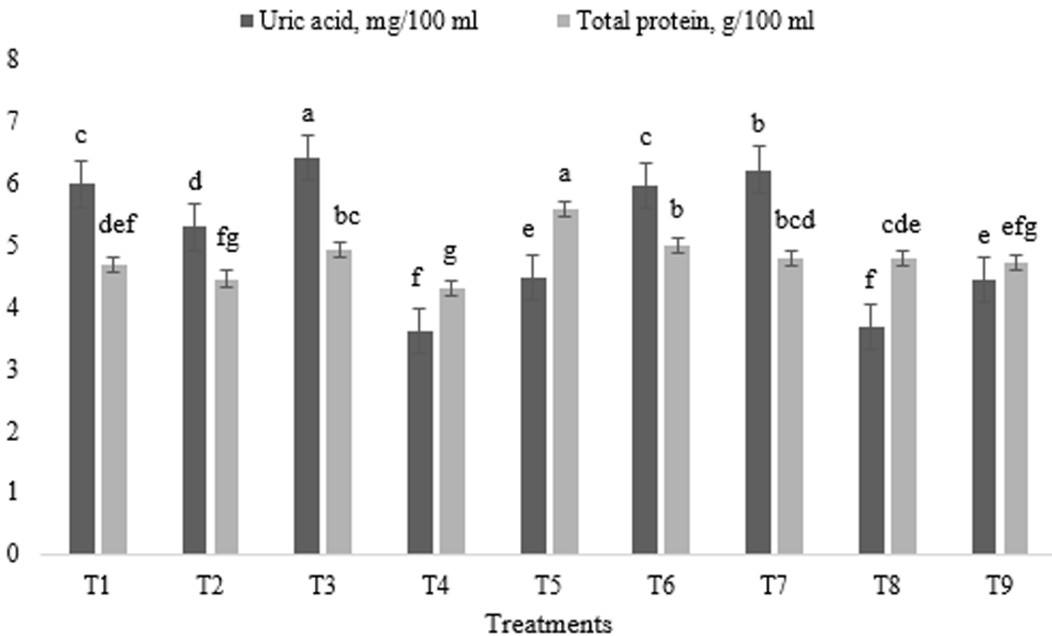

**Fig 2. Impact of astaxanthin and artichoke leaf powder on broiler blood uric acid and total protein concentrations.** Treatments: T1 = Control treatment; T2 & T3 = 60 and 120 mg astaxanthin/ kg feed; T4 & T5 = 1 and 2 g artichoke leaf powder/ kg feed; T6 = 60 mg astaxanthin + 1 g artichoke leaf powder)/ kg feed; T7 = 60 mg astaxanthin + 2 g artichoke leaf powder/ kg feed; T8 = 120 mg astaxanthin + 1 g artichoke leaf powder/ kg feed; T9 = 120 mg astaxanthin + 2 g artichoke leaf powder/ kg feed. Means within a column marked with different superscripts indicate statistically significant differences $p < 0.01$ or $p < 0.05$.

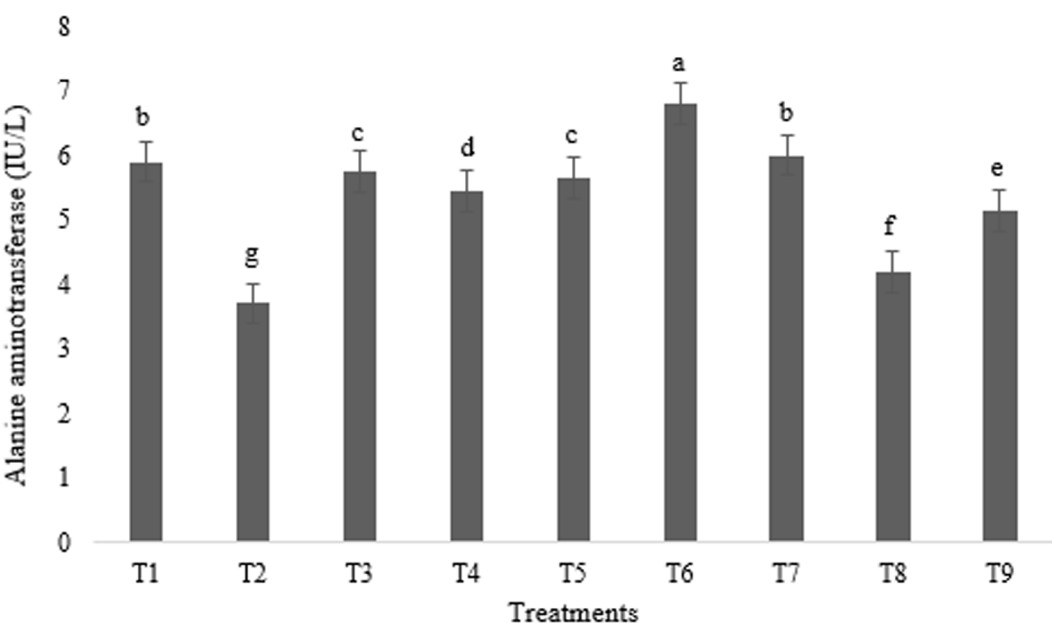

**Fig 3. Impact of astaxanthin and artichoke leaf powder on alanine aminotransferase enzyme activities in broiler chickens (IU/L).** Treatments: T1 = Control treatment; T2 & T3 = 60 and 120 mg astaxanthin/ kg feed; T4 & T5 = 1 and 2 g artichoke leaf powder/ kg feed; T6 = 60 mg astaxanthin + 1 g artichoke leaf powder)/ kg feed; T7 = 60 mg astaxanthin + 2 g artichoke leaf powder/ kg feed; T8 = 120 mg astaxanthin + 1 g artichoke leaf powder/ kg feed; T9 = 120 mg astaxanthin + 2 g artichoke leaf powder/ kg feed. Means within a column marked with different superscripts indicate statistically significant differences $p < 0.01$ or $p < 0.05$.

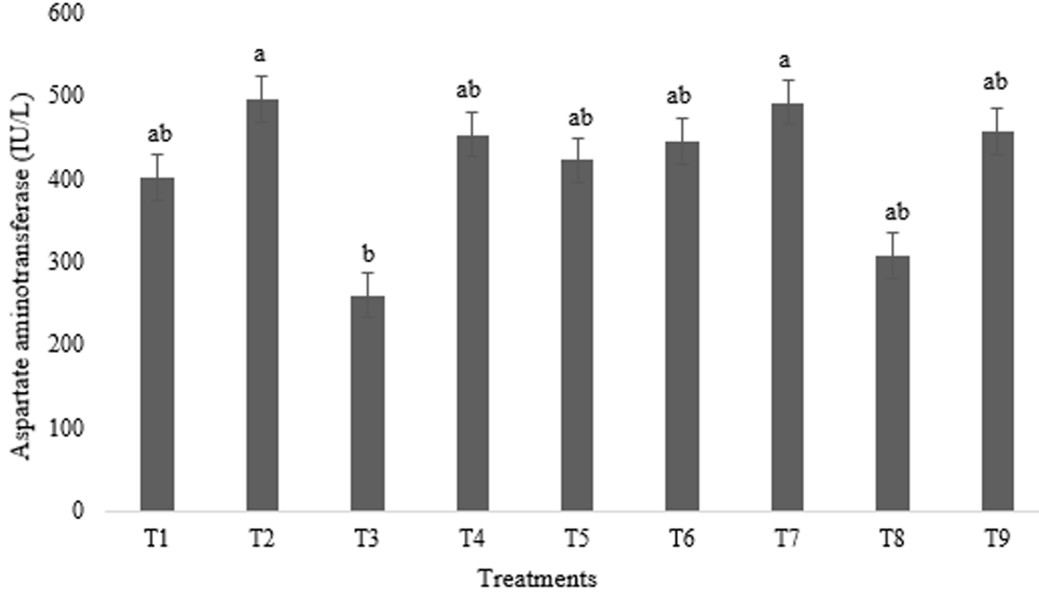

**Fig 4. Impact of astaxanthin and artichoke leaf powder on aspartate aminotransferase enzyme activities in broiler chickens (IU/L).** Treatments: T1 = Control treatment; T2 & T3 = 60 and 120 mg astaxanthin/ kg feed; T4 & T5 = 1 and 2 g artichoke leaf powder/ kg feed; T6 = 60 mg astaxanthin + 1 g artichoke leaf powder)/ kg feed; T7 = 60 mg astaxanthin + 2 g artichoke leaf powder/ kg feed; T8 = 120 mg astaxanthin + 1 g artichoke leaf powder/ kg feed; T9 = 120 mg astaxanthin + 2 g artichoke leaf powder/ kg feed. Means within a column marked with different superscripts indicate statistically significant differences $p < 0.01$ or $p < 0.05$.

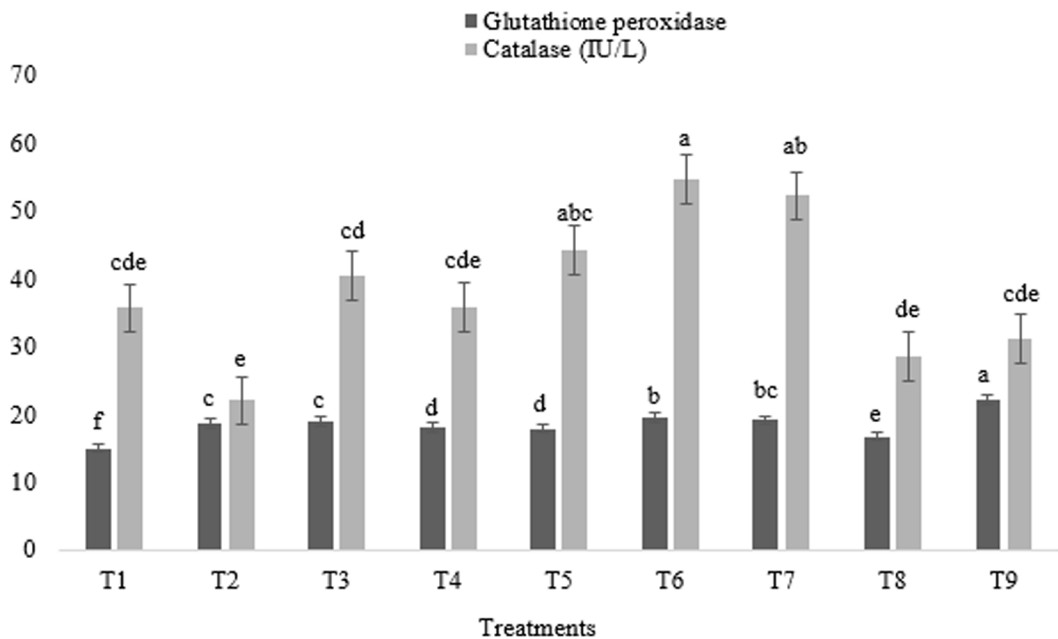

**Fig 5. Impact of astaxanthin and artichoke leaf powder on antioxidant enzyme activities in broiler chickens (IU/L).** Treatments: T1 = Control treatment; T2 & T3 = 60 and 120 mg astaxanthin/ kg feed; T4 & T5 = 1 and 2 g artichoke leaf powder/ kg feed; T6 = 60 mg astaxanthin + 1 g artichoke leaf powder)/ kg feed; T7 = 60 mg astaxanthin + 2 g artichoke leaf powder/ kg feed; T8 = 120 mg astaxanthin + 1 g artichoke leaf powder/ kg feed; T9 = 120 mg astaxanthin + 2 g artichoke leaf powder/ kg feed. Means within a column marked with different superscripts indicate statistically significant differences $p < 0.01$ or $p < 0.05$.

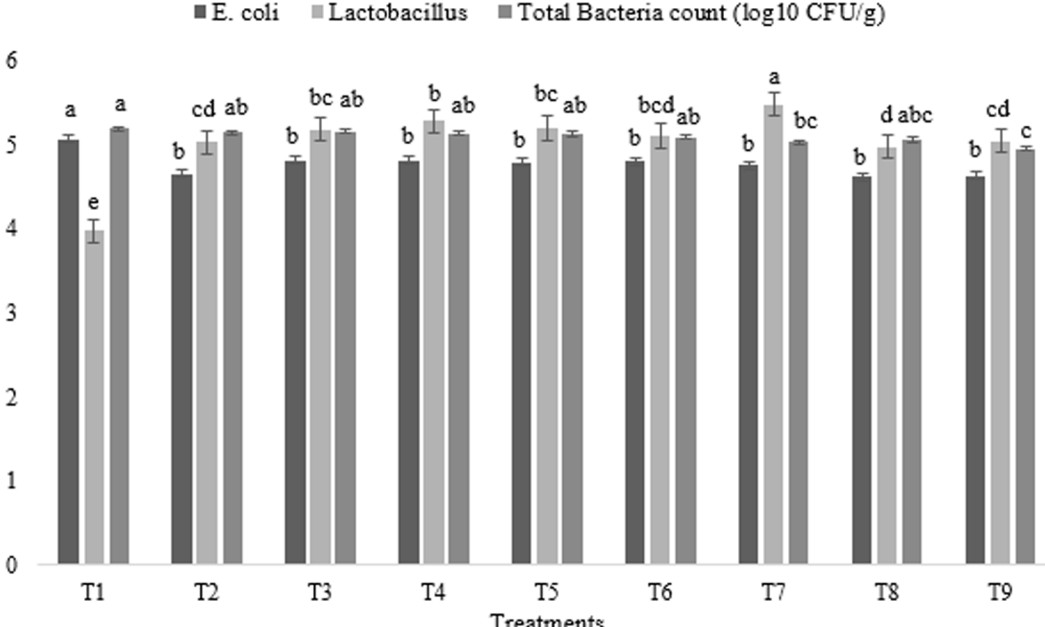

**Fig 6. Impact of astaxanthin and artichoke leaf powder on cecal microbial populations in broiler chickens (log10 CFU/g).** Treatments: T1 = Control treatment; T2 & T3 = 60 and 120 mg astaxanthin/ kg feed; T4 & T5 = 1 and 2 g artichoke leaf powder/ kg feed; T6 = 60 mg astaxanthin + 1 g artichoke leaf powder)/ kg feed; T7 = 60 mg astaxanthin + 2 g artichoke leaf powder/ kg feed; T8 = 120 mg astaxanthin + 1 g artichoke leaf powder/ kg feed; T9 = 120 mg astaxanthin + 2 g artichoke leaf powder/ kg feed. Means within a column marked with different superscripts indicate statistically significant differences $p < 0.01$ or $p < 0.05$.

was observed in all treated groups relative to the control (p < 0.05). Conversely, *Lactobacillus* populations were significantly lower in the control group and exhibited a substantial increase in all treated groups, with broilers under T7 demonstrating the highest levels (p < 0.01). Furthermore, Total bacterial counts were highest in the control group, significantly exceeding those observed in broilers receiving T7 and T9, which displayed the lowest counts (p < 0.05).

## Discussion

This study aimed to investigate the effects of dietary astaxanthin and ALP, both individually and in combination, on growth performance, hematological parameters, blood biochemical indices, enzyme activity, and cecal microbiota composition in broiler chickens, ultimately demonstrating positive impacts across a range of physiological and performance metrics.

The significant variations in LBW and BWG observed across treatment groups clearly illustrate the growth-modulating potential of dietary astaxanthin and ALP. A dose-dependent BWG increase, peaking in broilers fed T2 and T6, highlights their efficacy, aligning with their documented roles in nutrient absorption, gut health, and immune modulation [4,10]. Astaxanthin, a potent antioxidant, effectively mitigates oxidative stress, thereby fostering optimal growth conditions [19]. Conversely, ALP, rich in bioactive compounds, is known to enhance digestive enzyme activity and improve liver function, leading to more efficient nutrient utilization [9]. The consistently lower LBW and BWG observed in the control group underscores the beneficial impact of these supplemented diets. The findings are consistent with a substantial body of literature demonstrating the positive effects of dietary antioxidants and functional ingredients on poultry growth. For example, some studies reported improved BWG in broilers supplemented with astaxanthin, validating its growth-promoting properties [6,20]. However, conflicting results, such as those reported by Takahashi et al [21], who found no significant effects of astaxanthin on BWG, emphasize the need for careful consideration of experimental parameters. Similarly, the effects of ALP on growth performance have shown variability. A study demonstrated significant increases in BWG with artichoke premix concentrate [12], whereas another investigation found no significant effects with artichoke extract [22]. Interestingly, a research study reported decreased LBW and BWG with ALP, a finding that partially aligns with the reduced growth observed in broilers receiving T4 [23]. This potential negative impact could be attributed to the high soluble fiber content of ALP, including inulin and pectin, which may increase digesta viscosity and hinder nutrient absorption [9,10]. Factors such as experimental design, dietary composition, the source and dosage of astaxanthin and ALP, broiler strain, and the potential interactions between dietary components likely contribute to these variations in different studies.

The variations in FI across the experimental treatments suggest that astaxanthin and ALP can influence the palatability and appetite of broiler chickens. Elevated FI in the groups receiving T4 and T2 indicates ALP's potential to stimulate appetite, possibly via polyphenols and inulin enhancing palatability and digestion [9]. Astaxanthin may counteract oxidative stress-induced appetite suppression [4], thereby influencing FI, albeit with dose-dependent effects and potential interactions with ALP. However, high FI did not always translate to better growth performance, as evidenced by the high FCR observed in broilers under T4. Conversely, the groups receiving T6 and T8 consistently demonstrated the lowest FCR, indicating that these treatments optimized the conversion of feed to body weight. This improvement in FCR could be attributed to the combined effects of astaxanthin and ALP on gut health, nutrient digestibility, and metabolic efficiency [4,10]. The control group's poor FCR underscores the importance of dietary supplementation for improving broiler performance. Conflicting findings have been reported in the literature regarding the effects of astaxanthin and ALP on FI and FCR. While a study observed no significant effect of astaxanthin on FI [21], and a research found no differences in FCR [20], positive correlations were noted in other investigations [7,19]. Similarly, an experiment reported improved FCR with astaxanthin-rich red yeast [6]. Regarding ALP, no effects were observed in certain studies [22,24], whereas an increase in FI and a decrease in FCR were documented in another research [11].

Hematological indices serve as critical indicators of an animal's physiological status, reflecting the intricate interplay between nutritional inputs and overall health [25]. While RBC counts, PCV, and Hb concentration showed some differences, trends were nuanced. Broilers given T4 and T8, despite initially appearing to have higher RBC counts, were not

statistically different from the control (T1). This suggests that ALP dietary inclusion, while potentially influencing erythropoiesis, did not significantly alter RBC counts at the tested concentrations, especially in combination with astaxanthin (T8). Similarly, although broilers receiving T9 exhibited the highest PCV and Hb values, and those on T7 showed the lowest, these differencesdid not achieve statistical significance when compared to T1 or other treatments. The observed trends are likely attributable to the combined antioxidant properties of astaxanthin and ALP-derived polyphenols [4,10]. Astaxanthin is known to protect cellular membranes from oxidative stress [4,16] and enhance endogenous antioxidant enzyme activity [26], potentially stimulating erythropoiesis [27]. ALP, rich in phenolic compounds [8], complements these effects by scavenging free radicals and mitigating oxidative damage. We hypothesized a synergistic interaction between astaxanthin and ALP, potentially leading to enhanced erythrocyte health. However, the absence of consistent, statistically significant improvements across all treatments necessitates a nuanced interpretation. Further research is warranted to optimize the ratios and concentrations of these additives to maximize their potential benefits. These findings partially align with previous research. While some studies have shown improved hematological parameters with ALP [28] or astaxanthin [29], our results indicate inconsistent significant improvements. This discrepancy may be attributed to variations in experimental design, animal models, supplement sources, and dosage regimes. Similar to Al-Kassie and Al-Qaraghuli [30], we observed potential benefits with polyphenols, but the compounds and dosages differed. Critically, the findings also echo studies reporting no significant RBC effects with lower astaxanthin doses (e.g., 10–20 g/kg) [6].

Although total WBC counts remained stable across treatments, significant shifts in leukocyte subtypes revealed distinct immunomodulatory effects. The group receiving T8 significantly elevated heterophil percentages and the H/L ratio, suggesting a potential stress response. Conversely, broilers provided with T2, T6 and T9 exhibited significantly reduced heterophil percentages and H/L ratios, indicating a potential stress-mitigating effect. This likely reflects the antioxidant properties of astaxanthin and the bioactive compounds in ALP. Astaxanthin, a potent carotenoid, is known to scavenge free radicals and modulate inflammatory pathways [4]. Similarly, ALP, rich in phenolic compounds like cynarin and flavonoids, has demonstrated capacity to reduce oxidative stress and systemic inflammation [9,24]. Moreover, artichoke's significant vitamin C content, comparable to common fruits and vegetables, bolsters immune function by supporting cellular immune processes [9]. The observed reduction in lymphocyte counts in broilers given T4 represents a critical finding that warrants further investigation. Lymphocytes are pivotal components of the adaptive immune system, and their decline may impact specific immune functions, such as antibody production and cell-mediated immunity. Further scrutiny is needed to determine the underlying mechanisms and potential long-term implications for immune competence. Previous studies have indicated that phenolic compounds in ALP can modulate immune responses, though their effects are dose-dependent and influenced by interactions with other dietary components [9]. Shevchenko et al [31] reported reduced leukocyte levels with higher astaxanthin doses (20–30 mg/kg), while lower doses (10 mg/kg) had no effect. Similarly, Jeong and Kim [6] suggested that astaxanthin concentrations from fermented *P. rhodozyma* at 2.3 and 4.6 mg/kg may be insufficient for robust immune stimulation compared to higher doses (100 mg/kg). The findings, utilizing 60 and 120 mg/kg astaxanthin, contribute to this body of literature by demonstrating dose-dependent immunomodulatory effects.

In this study, all treatments, with the exception of the group receiving T4, elicited a significant elevation in glucose levels compared to the control group. The observed hyperglycemic response following astaxanthin supplementation aligns with findings from a study in which a dose-dependent increase in serum glucose was reported in laying hens administered 10, 20, or 30 mg/kg astaxanthin [32]. However, these results differ from previous research which demonstrated hypoglycemic effects of astaxanthin, attributed to its potential to stimulate insulin production [33]. Moreover, ALP did not significantly affect blood glucose concentration in broiler chickens [34]. This discrepancy may be attributed to several factors, including differences in species, dosage, duration of treatment, and the overall dietary composition. The hyperglycemia observed with astaxanthin is likely due to its enhancement of mitochondrial function, which can stimulate glucose production through gluconeogenesis or glycogenolysis [35,36]. Additionally, the increased mitochondrial activity may lead to a transient rise in glucose availability to support the augmented ATP production [36].

Astaxanthin supplementation exhibited a dose-dependent effect on cholesterol. While lower doses showed no significant change, higher doses (T3 and T8) increased cholesterol, suggesting a potential modulation of cholesterol metabolism. Further research is needed to pinpoint the mechanisms. Conversely, ALP supplementation, particularly at 1g/kg (the T4 group), significantly reduced cholesterol, indicating its hypolipidemic potential. The astaxanthin results of this study differ from studies reporting no effect [26,37], and those showing lipid lowering effects [4], highlighting variability likely due to experimental differences. The cholesterol-lowering by ALP aligns with findings on natural hypolipidemic agents. Artichoke studies [12,28,34] show similar reductions, attributed to cynarin's HMG-CoA reductase inhibition [9].

ALP supplementation (the T4 and T5 groups) significantly reduced uric acid, potentially through xanthine oxidase inhibition, enhanced liver and kidney function, antioxidant activity, and prebiotic effects [9,38]. These mechanisms likely improve uric acid management, though further research is needed to define ALP's direct impact on poultry uric acid metabolism. Astaxanthin's effect was dose-dependent: a trend towards reduction with lower doses (T2) and a significant increase with higher doses (T3), suggesting a biphasic response that warrants further investigation. Notably, combined astaxanthin and ALP treatments (the T6-T9 groups) showed lower uric acid than the control, indicating a potential interaction. The dominant effect of ALP in these combinations may have mitigated the hyperuricemic effect of high-dose astaxanthin alone, highlighting the importance of combination therapies and their metabolic interactions. The current results differ from those reported in a study [37], in which astaxanthin-induced uric acid reduction was observed, and from another investigation, which found no effect of ALP [34]. These discrepancies likely stem from variations in species, experimental design, AST dosage, ALP dosage, and treatment duration.

ALP supplementation exhibited a dose-dependent increase in total protein, suggesting a direct influence on protein metabolism. Astaxanthin's impact was less consistent: a lower dose decreased protein, while a higher dose showed no significant change. Notably, combined astaxanthin and ALP treatments displayed elevated protein levels, indicating a potential synergistic interaction that mitigated the protein-reducing effect of low-dose astaxanthin alone. This synergy might result from enhanced nutrient assimilation or reduced protein oxidative damage, implying a role for ALP and astaxanthin in promoting protein synthesis or inhibiting degradation [4,9]. Previous research presents a mixed picture. Fallah et al [34] reported no effect of 1.5% ALP on blood total protein of broilers. Conversely, astaxanthin at 50 and 100 mg/kg significantly increased broiler total protein [37], potentially due to its antioxidant properties. Astaxanthin may enhance protein synthesis by suppressing corticosterone, a catabolic hormone, and stimulating lymphocyte production [26,33]. These discrepancies highlight the context-dependent nature of these compounds' effects, necessitating further mechanistic studies.

AST and ALT are established biomarkers for assessing hepatic integrity, with their release into circulation typically indicative of hepatocyte damage [39]. In this study, individual administration of astaxanthin and ALP significantly reduced ALT activity compared to the control group, consistent with reported hepatoprotective effects [37,40]. The ALT-lowering effect of astaxanthin is likely attributed to its potent antioxidant capacity, mediated by its polyene chain, which effectively scavenges free radicals and preserves cell membrane integrity [37]. Similarly, ALP has demonstrated hepatoprotective potential in lead-induced hepatotoxicity models, reducing both ALT and AST levels [40]. This effect is likely due to its antioxidant and anti-inflammatory properties of phenolic compounds such as caffeic acid and isochlorogenic acid, which mitigate liver damage and maintain cellular integrity [9]. However, a significant increase in ALT activity was observed in broiler fed T6. This unexpected synergistic increase suggests a potential antagonistic interaction at this specific dosage ratio, necessitating further investigation to elucidate the underlying mechanisms. Regarding AST, no statistically significant differences were observed across treatments, although the lowest enzyme concentration was noted in the groups receiving T3. This result contrasts with some studies demonstrating significant effects of artichoke on AST and ALT [41]. However, it aligns with reports indicating no significant impact of ALP on broiler blood AST and ALT [23,28]. These discrepancies may stem from variations in experimental design, animal models, or the specific compounds and dosages employed.

All treatments significantly elevated GPX, with broilers under T9 showing the highest levels, indicating a potent synergistic antioxidant effect, consistent with astaxanthin and ALP's established ROS-scavenging abilities [37,42]. GPX, a

crucial antioxidant enzyme, mitigates oxidative stress by catalyzing the reduction of hydrogen peroxide, thus neutralizing ROS [2]. The findings corroborate reports of increased GPX activity with astaxanthin supplementation in broilers, attributed to improved cellular antioxidant status [37]. Similarly, astaxanthin from *Haematococcus pluvialis* has been shown to elevate GPX activity, albeit with variations depending on supplementation duration [42]. Combined 60 mg astaxanthin with ALP (the T6 and T7 groups) also enhanced CAT activity, further supporting improved antioxidant capacity, likely through the unique structure of astaxanthin, with its polar rings and polyene chain [4]. This is consistent with observations of astaxanthin-induced increases in plasma CAT levels [20]. CAT, like GPX, plays a critical role in ROS detoxification [2]. While some studies report decreased CAT/GPX with ALP [43], others confirm its antioxidant efficacy via increased CAT and GSX [44,45], attributed to polyphenol-mediated ROS scavenging and lipid peroxidation inhibition [9].

Astaxanthin and ALP significantly altered broiler cecal microbiota, notably reducing *E. coli* and increasing *Lactobacillus* abundance. Consistent *E. coli* reduction across treatments likely results from astaxanthin's antioxidant-mediated gut barrier enhancement, limiting pathogen colonization, and ALP's direct antimicrobial polyphenols [4,46]. Astaxanthin's potential to bolster tight junction proteins and reduce pro-inflammatory cytokines, as detailed by Gao et al [47] further supports this. Increased *Lactobacillus* abundance reflects ALP's prebiotic inulin/ fructooligosaccharides content, synergistically enhanced by astaxanthin's protective effects on probiotic survival [4,48]. The lowest total bacterial counts in high-ALP combinations (broilers in the T7 and T9 groups) suggest selective microbial restructuring, favoring *Lactobacillus* and fermentative taxa via reduced nutrient availability for non-specialist bacteria [49]. However, a plateau effect in astaxanthin efficacy above 60 mg/kg indicates limited absorption, emphasizing balanced formulation [4,42].

## Conclusion

This study demonstrates that specific dietary combinations of astaxanthin and ALP significantly enhance broiler performance and health. The groups fed T2 and T6 improved growth performance, exhibiting increased LBW, BWG, and an improved FCR, indicating better nutrient utilization. Several treated groups, particularly those receiving T2, T6, and T9, mitigated stress by reducing heterophil and H/L ratios. Metabolically, the group receiving T4 effectively lowered cholesterol, while the groups fed T4, T5, T8, and T9 reduced uric acid levels. Most supplemented groups, especially the T9 group, showed increased antioxidant capacity (demonstrated by higher GPX activity), and the T6 and T7 groups showed increased CAT activity. Crucially, all supplemented treatment groups improved gut health by reducing *E. coli* populations and increasing *Lactobacillus* counts in the cecum. In conclusion, dietary astaxanthin and ALP, especially for the group fed T6, can boost broiler production efficiency and overall health. However, further research into carcass characteristics and meat quality in broilers is recommended.

## Supporting information

**S1 Data. The excel file of performance data is presented as S1 Data.**
(XLSX)

## Acknowledgments

The authors appreciate the College of Agriculture and Natural Resources, Razi University, Iran and College of Agriculture, Al-Qasim Green University, Iraq.

## Author contributions

**Conceptualization:** Mehran Torki, Fadhil Rasoul Abbas Al-Khafaji.
**Data curation:** Saad Naji Nasser, Fadhil Rasoul Abbas Al-Khafaji, Ali Ahmad Alaw Qotbi.
**Formal analysis:** Saad Naji Nasser, Fadhil Rasoul Abbas Al-Khafaji, Ali Ahmad Alaw Qotbi.

**Investigation:** Saad Naji Nasser, Mehran Torki, Fadhil Rasoul Abbas Al-Khafaji.

**Methodology:** Saad Naji Nasser, Mehran Torki, Fadhil Rasoul Abbas Al-Khafaji, Shahab Ghazi Harsini.

**Project administration:** Mehran Torki, Fadhil Rasoul Abbas Al-Khafaji.

**Resources:** Mehran Torki, Fadhil Rasoul Abbas Al-Khafaji.

**Software:** Saad Naji Nasser, Shahab Ghazi Harsini, Ali Ahmad Alaw Qotbi.

**Supervision:** Mehran Torki, Fadhil Rasoul Abbas Al-Khafaji.

**Validation:** Mehran Torki, Fadhil Rasoul Abbas Al-Khafaji, Shahab Ghazi Harsini, Ali Ahmad Alaw Qotbi.

**Writing – original draft:** Saad Naji Nasser.

**Writing – review & editing:** Mehran Torki, Fadhil Rasoul Abbas Al-Khafaji, Shahab Ghazi Harsini, Ali Ahmad Alaw Qotbi.

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
