## [Decision Letter · Decision Letter 0]

23 May 2025

Dear Dr. Torki,

Thank you for submitting your manuscript to PLOS ONE. After careful consideration, we feel that it has merit but does not fully meet PLOS ONE’s publication criteria as it currently stands. Therefore, we invite you to submit a revised version of the manuscript that addresses the points raised during the review process.

We look forward to receiving your revised manuscript.

Kind regards,

Katrien G. Janin, PhD,

Staff Editor,

on behalf of

Sameh Abdelnour

Academic Editor

PLOS ONE

Journal Requirements:

3. Please include a complete copy of PLOS’ questionnaire on inclusivity in global research in your revised manuscript. Our policy for research in this area aims to improve transparency in the reporting of research performed outside of researchers’ own country or community. The policy applies to researchers who have travelled to a different country to conduct research, research with Indigenous populations or their lands, and research on cultural artefacts. The questionnaire can also be requested at the journal’s discretion for any other submissions, even if these conditions are not met.  Please find more information on the policy and a link to download a blank copy of the questionnaire here: https://journals.plos.org/plosone/s/best-practices-in-research-reporting. Please upload a completed version of your questionnaire as Supporting Information when you resubmit your manuscript.

**Additional Editor Comments:**

Dear Dr. Mehran Torki

Have a good day

The reviewer suggests revisions to your manuscript according to their feedback. Please carefully address all comments when you submit your revised manuscript.

Reviewers' comments:

Reviewer's Responses to Questions

**Comments to the Author**

1. Is the manuscript technically sound, and do the data support the conclusions?

Reviewer #1: Yes

Reviewer #2: Yes

2. Has the statistical analysis been performed appropriately and rigorously?

Reviewer #1: Yes

Reviewer #2: Yes

3. Have the authors made all data underlying the findings in their manuscript fully available?

Reviewer #1: Yes

Reviewer #2: Yes

4. Is the manuscript presented in an intelligible fashion and written in standard English?

Reviewer #1: Yes

Reviewer #2: Yes

Reviewer #1: I have reviewed the manuscript thoroughly, and found it to be well-written and comprehensive. The authors have presented their findings clearly and effectively. The study's methodology is sound, and the results are well-supported by the data provided.

The manuscript offers valuable insights into the effects of dietary supplementation with artichoke leaf powder and astaxanthin on broiler chickens, covering various aspects such as productive performance, hematological and biochemical profiles, hepatic and antioxidant enzyme activities, and gut microbiota. The discussion is thorough and provides a good interpretation of the results in the context of existing literature.

I have made my comments and amendments using the Track Changes feature in the manuscript for clarity and ease of review.

Overall, I believe this manuscript makes a significant contribution to the field and would be a valuable addition to the journal.

Reviewer #2: Title: Diet inclusion of artichoke (Cynara scolymus) leaf powder and astaxanthin and evaluating productive performance of broiler chickens, hematological and biochemical profiles, hepatic and antioxidant enzyme activities, and gut microbiota

General Comments:

The present study offers a compelling investigation into the synergistic effects of astaxanthin and artichoke leaf powder (ALP) on broiler chickens, with significant implications for poultry nutrition and health. By evaluating growth performance, hematological and biochemical parameters, hepatic and antioxidant enzyme activities, and gut microbiota, this research provides valuable insights into the potential of these natural supplements to enhance broiler productivity and metabolic health. The findings demonstrate that supplementation with 60 mg astaxanthin + 1 g ALP significantly improves growth performance, antioxidant status, lipid/uric acid metabolism, and gut microbial composition.

Recommendation: Minor revision

1. Abstract:

Add a brief introductory sentence to establish the relevance and benefits of natural feed additives in poultry production before presenting the study’s objectives.

2. Introduction:

The introduction is well-structured, providing a solid background on broiler production challenges and the potential benefits of astaxanthin and ALP.

3. Materials and Methods:

Overall, this section is clearly written and provides sufficient methodological detail. The statistical approaches are appropriate and well applied.

4. Results:

- Tables 2, 3, 4, and 5: Consider consolidating these tables into a single comprehensive table showing growth performance variables (initial weight, final weight, body weight gain, feed intake, feed conversion ratio) across all experimental periods. This would enhance readability and allow for easier comparison of treatment effects over time.

- Tables 7, 8, and 9: It is recommended to convert these into graphical representations (bar or line graphs) to visually emphasize the trends and significant differences among treatment groups.

- Ensure that the narrative of the Results section is updated accordingly to reflect the restructured tables and figures.

5. Discussion:

The discussion is logically organized and provides a solid interpretation of the findings in the context of previous research.

6. Conclusion:

Please revise this section for brevity and clarity. A more concise summary that highlights the most significant findings and practical implications of the study would be more impactful.

**Do you want your identity to be public for this peer review?** For information about this choice, including consent withdrawal, please see our Privacy Policy

Reviewer #1: No

Reviewer #2: No

---

## [Author Response · Author response to Decision Letter 1]

1 Jul 2025

The file of Response to Reviewers has been also uploaded.

Reviewer #1:

I have reviewed the manuscript thoroughly, and found it to be well-written and comprehensive. The authors have presented their findings clearly and effectively. The study's methodology is sound, and the results are well-supported by the data provided. The manuscript offers valuable insights into the effects of dietary supplementation with artichoke leaf powder and astaxanthin on broiler chickens, covering various aspects such as productive performance, hematological and biochemical profiles, hepatic and antioxidant enzyme activities, and gut microbiota. The discussion is thorough and provides a good interpretation of the results in the context of existing literature. I have made my comments and amendments using the Track Changes feature in the manuscript for clarity and ease of review. Overall, I believe this manuscript makes a significant contribution to the field and would be a valuable addition to the journal.

Thank you for your thorough review and positive assessment of our manuscript. We truly appreciate you taking the time to provide such detailed comments and amendments via Track Changes. We want to assure you that we've carefully addressed every one of your suggestions and implemented all the indicated changes throughout the entire manuscript, which we believe has significantly enhanced its clarity and quality.

How many replicates in each dietary treatment?

Thank you for your question. Six replicates per treatment, 20 birds per replicate

Instead of stating "T4 showed..." or "T2 exhibited...", I recommend using phrases that specify the group receiving the treatment. For example, you could say, "The group that received T3 showed higher BWG..." or "The group fed T2 exhibited the greatest increase in BWG...". This approach will make it clearer which groups are being referred to and improve the overall flow of your writing. Check the whole manuscript

Thank you for this valuable suggestion. We have carefully reviewed and implemented this change throughout the entire manuscript.

In this line and elsewhere: To maintain a formal and objective tone, it would be beneficial to avoid using personal pronouns such as "our findings" or "we found." Instead, you could use phrases like "The findings indicate..." or "It was observed that...

Thank you for this valuable feedback. This change has been applied consistently to enhance the scientific impartiality of the writing.

Reviewer #2:

The present study offers a compelling investigation into the synergistic effects of astaxanthin and artichoke leaf powder (ALP) on broiler chickens, with significant implications for poultry nutrition and health. By evaluating growth performance, hematological and biochemical parameters, hepatic and antioxidant enzyme activities, and gut microbiota, this research provides valuable insights into the potential of these natural supplements to enhance broiler productivity and metabolic health. The findings demonstrate that supplementation with 60 mg astaxanthin + 1 g ALP significantly improves growth performance, antioxidant status, lipid/uric acid metabolism, and gut microbial composition.

Recommendation: Minor revision

Thank you for your very positive assessment of our manuscript. We're delighted to hear that you found our investigation into the synergistic effects of astaxanthin and artichoke leaf powder (ALP) on broiler chickens compelling, and that you recognize its significant implications for poultry nutrition and health. We've addressed all the points raised in your review, and all the suggested changes have been implemented throughout the manuscript. We believe these revisions have significantly enhanced the clarity and quality of our work.

1. Abstract: Add a brief introductory sentence to establish the relevance and benefits of natural feed additives in poultry production before presenting the study’s objectives.

Thank you for your comment. We've added a brief introductory sentence to the abstract to introduce the relevance and benefits of natural feed additives in poultry production before diving into the study's objectives.

2. Introduction: The introduction is well-structured, providing a solid background on broiler production challenges and the potential benefits of astaxanthin and ALP.

Thank you for your positive feedback on our introduction.

3. Materials and Methods: Overall, this section is clearly written and provides sufficient methodological detail. The statistical approaches are appropriate and well applied.

Thank you for your positive assessment of our materials and methods section.

4. Results: - Tables 2, 3, 4, and 5: Consider consolidating these tables into a single comprehensive table showing growth performance variables (initial weight, final weight, body weight gain, feed intake, feed conversion ratio) across all experimental periods. This would enhance readability and allow for easier comparison of treatment effects over time.

Thank you for your suggestion. We’ve consolidated Tables 2, 3, 4, and 5 into a single, comprehensive Table 2. This new table now presents all growth performance variables—initial weight, final weight, body weight gain, feed intake, and feed conversion ratio—across all experimental periods.

- Tables 7, 8, and 9: It is recommended to convert these into graphical representations (bar or line graphs) to visually emphasize the trends and significant differences among treatment groups.

- Ensure that the narrative of the Results section is updated accordingly to reflect the restructured tables and figures.

Thank you for your suggestion. We've converted Tables 7, 8, and 9 into bar graphs (Figures 1, 2, 3, 4, 5, and 6) to visually emphasize the trends and significant differences among treatment groups. We've also updated the narrative of the results section to accurately reflect these new figures and the restructured information.

5. Discussion: The discussion is logically organized and provides a solid interpretation of the findings in the context of previous research.

Thank you for your positive feedback on our discussion section.

6. Conclusion: Please revise this section for brevity and clarity. A more concise summary that highlights the most significant findings and practical implications of the study would be more impactful.

We've revised the conclusion section for brevity and clarity. The updated section now offers a more concise summary, highlighting only the most significant findings and the practical implications of our study to enhance its impact.

---

## [Editor Report · Decision Letter 1]

8 Jul 2025

Effect of dietary inclusion of artichoke (Cynara scolymus) leaf powder and astaxanthin on productive performance, hematological and biochemical profiles, hepatic and antioxidant enzyme activities, and gut microbiota in broiler chickens

PONE-D-25-20943R1

Dear Dr. Mehran Torki,

We’re pleased to inform you that your manuscript has been judged scientifically suitable for publication and will be formally accepted for publication once it meets all outstanding technical requirements.

Kind regards,

Sameh Abdelnour

Academic Editor

PLOS ONE

Additional Editor Comments (optional):

Authors addressed all the comments.
---

## [Editor Report · Acceptance letter]

PONE-D-25-20943R1

PLOS ONE

Dear Dr. Torki,

I'm pleased to inform you that your manuscript has been deemed suitable for publication in PLOS ONE. Congratulations! Your manuscript is now being handed over to our production team.

Kind regards,

on behalf of

Dr. Sameh Abdelnour

Academic Editor

PLOS ONE